# The Moderating Effect of Distance on Features of the Built Environment and Active School Transport

**DOI:** 10.3390/ijerph17217856

**Published:** 2020-10-27

**Authors:** Allison Ross, Josephine Godwyll, Marc Adams

**Affiliations:** College of Health Solutions, Arizona State University, Phoenix, AZ 85004, USA; jmgodwyl@asu.edu (J.G.); marc.adams@asu.edu (M.A.)

**Keywords:** active school transport, built environment, walking and biking to school

## Abstract

Despite growing research supporting the impact of the built environment on active school transport (AST), distance persists as the most powerful predictor of walking and biking to school. There is a need to better understand how environmental features interact with distance to affect AST, and whether the influence of environmental factors persist across different distance thresholds. Multilevel models using cluster-robust standard errors were used to examine for interactions between objectively measured macroscale environmental features and several reported distances from home to school (up to ¼, ¼ up to ½, ½ up to 1, 1+ miles) on the likelihood of parent reported AST for children grades 3–8 (*n* = 2751) at 35 schools who completed a Safe Routes to School Parent Survey about Walking and Biking to School (SRTS Parent Survey). An interaction between both intersection density and food-related land use with distance was observed. The likelihood of AST decreased as intersection density and distance increased (i.e., 31.0% reduced odds among those living within ¼ mile compared to 18.2% using ½–1-mile criterion). The likelihood of using AST were reduced as food-related land use and distance increased (i.e., 43.67% reduced odds among those living under ¼ mile compared to 19.83% reduced odds among those living ½–1 mile). Programs and infrastructure improvements focused on overcoming environmental barriers to promote AST may be most effective when targeting neighborhoods within ¼ mile of schools.

## 1. Introduction

Physical activity during childhood may be a protective factor against chronic disease as children age [1]. Promoting active school transport (AST), or active travel to and from school such as walking or biking, has become a recognized public health priority as it represents a strategy to meet physical activity recommendations for children and adolescents [2,3,4]. AST is associated with a number of health benefits including improved cardiorespiratory fitness and body composition [5], and is related to higher cognitive performance [6]. Despite these health benefits, the United States (U.S.) has one of the lowest rates of walking and biking for transport [7] with only an estimated 10.4% of students using active forms of transportation to school [8], contributing to a “D-” grade for active transportation in the 2018 United States Report Card on Physical Activity for Children and Youth [9]. Considering the many health benefits of using active modes of school transport, encouraging and increasing AST has been emphasized as a key policy recommendation in the United States and internationally [10].

Features of the built environment are important determinants of physical activity [3,11,12] and AST [13,14]. Evidence of the specific impact of the built environment on physical activity and transport behavior is relatively consistent among adults [15,16,17] but less conclusive in studies of children’s school transport [14,18]. Commonly cited environmental correlates of AST include macroscale features such as density [19,20,21,22], route directness [14,23], and major street crossings [24,25], but the strongest and most consistently referenced determinant of AST is distance [14,26,27,28,29,30,31,32]. The impact of distance between home and school may be so powerful that, after a certain distance, it suppresses the contribution of elements of the built environment on children’s active travel [32].

Most studies investigating AST behaviors control for distance between home and school using a one-mile cutoff point (e.g., [33]). This distance criterion may be too general and fail to account for variability that exists within the range of one mile, particularly with regard to the effect of built environmental features. While some research has determined criterion distances other than the standard one mile, these cutoff points between active and non-active school transport range considerably (e.g., approximately 0.50 miles (800 m) [34,35], 0.62 miles (1 km) [36], 0.75 miles (1.3 km) [23], and 0.68–1.0 miles (1.1–1.6 kms) [27]). Given the need to determine established distance thresholds and consistent methodology around spatial relationships between measured environmental features and AST [14], we explored objectively measured features of the built environment and distance from home to school to determine which environmental features are associated with reported AST and whether relationships were moderated by different thresholds of distance.

## 2. Materials and Methods

### 2.1. Participants and Setting

The sample included parents of students (*n* = 2751) in grades 3–8 at 35 schools who completed a Safe Routes to School Parent Survey about Walking and Biking to School (SRTS Parent Survey) between 2010 and 2014 in the Greater Phoenix Metropolitan Area (Phoenix Metro, AZ, USA).

### 2.2. Research Design

The SRTS Parent Survey is a standard questionnaire used to gather information parent report of student school travel and parent perceptions of walking and biking to school [37]. Surveys are typically mailed or administered online to parents during SRTS programming. Parent responses are entered into a centralized data collection and reporting system supported by the Federal Highway Administration (FHWA). Data from surveys used in this research were obtained from this centralized system. Parent reports of the closest intersection to home were used as a proxy for home address to protect the anonymity of participants. These intersections were geocoded to identify surrounding built environment measures.

## 3. Measures

### 3.1. School Transport

Parents were asked to report their child’s mode of travel to and from school on most days. Responses were combined so “walk” and “bike” were collapsed to represent active travel and “school bus”, “family vehicle”, “carpool”, and “transit” were collapsed to represent inactive travel.

### 3.2. Built Environment

The entire Phoenix Metro region was represented as a large grid of 30 m × 30 m cells to develop a map of built environment features using Geographic Information Systems (GIS). Objective measures of built environment macroscale features previously related to AST were enumerated following existing methodologies [15]. Built environment features included intersection density (number of intersections with 3 segments or more), residential density (residential units per residential land area), transit density, and various land uses including residential, retail, office, recreational, food-related, entertainment, civic, and park categories during 2015–2016 [15]. Built environment features were summarized for each cell independently following the “Smartmap” strategy by Hurvitz et al. [38] using a 500-m Euclidean buffer. This geoprocessing process provided a unique value for each cell. Participant reported home location was geocoded and allocated to one previously described cell along with its associated built environment values.

### 3.3. Distance

Parents reported distance between home and school as either up to ¼ mile, ¼ up to ½ mile, ½ mile up to 1 mile, 1 mile up to 2 miles, or more than 2 miles. The latter two categories were combined to include one category representing all distances one mile or greater.

### 3.4. Covariates

Student gender, student grade, and parent level of education were included from the parent survey. Schools were placed into low-, medium-, and high-income categories determined by the percentage of students eligible to receive free and reduced priced meals (FRPM) using information from the Arizona Department of Education [39]. In keeping with work conducted by the National Center for Safe Routes to School [40], low-income schools were defined as having ≥75% of students eligible to receive FRPM; medium-income schools as having 40–75% of students eligible to receive FRPM; and high-income schools as having ≤40% of students eligible to receive FRPM. Using unique school identifiers and zip codes, we linked school-level information from parent survey data with National Center for Education Statistics-defined locale (a school’s proximity to an urbanized area, or region with a densely settled core with densely settled surrounding area) [41]. Schools were identified as city, suburb, or other.

### 3.5. Data Analysis

We included surveys from schools that entered data into the centralized system during 2010–2014 if: (1) they operated in the Phoenix Metro region; (2) they enrolled students in third through eighth grade; and (3) they entered more than 10 completed parent surveys each year the schools collected survey data. Built environment variables were collected during 2015–2016. Next, data were examined using descriptive statistics including means and standard deviations or percentages for covariates, predictor, and outcome variables and bivariate correlations with AST.

To account for the clustering of data by school site and possible violation of independence, intra-class correlation (ICC) coefficients were estimated to measure the amount of variance in walking and biking behavior by school. Because 16.5% of AST behavior was explained by differences among schools, multilevel modeling was utilized [42]. A series of logistic regression models was fit to identify characteristics of the home environment that were associated with the probability of students using active (walking or biking) versus inactive (riding in a bus or car) modes of transport to school. First, all environmental variables were entered into a single-level model to identify significant predictors of AST. We used a more liberal significance *p* < 0.10 level to reduce the likelihood of excluding relevant variables that may impact AST but only after adjusting for other covariates. Next, individual models with environmental variables and interaction terms were run to test for distance thresholds as moderators using interaction terms. Distance was dummy coded with ≥1 miles as the reference category. These models were adjusted for student gender and grade, parent level of education, and school-level income and locale. Because we were interested in the effect of environmental variables on AST at the individual level (and not interested in the extent those relationships vary by school site), we used a sandwich estimator with the CLUSTER and TYPE = COMPLEX command in Mplus to obtain corrected standard errors that account for school clustering [43]. Interactions between environmental correlates and distance were examined, and simple slopes among significant distance interactions were examined graphically. All independent variables were grand-mean centered. Thus, the interpretation of the fixed effects reflects the average change in the log-odds of walking and biking to school for a one-unit increase in each predictor across the overall sample, regardless of school cluster.

## 4. Results

The sample contained slightly more girls (53.7%) and more children in elementary (grades 3–5; 75%) compared to middle/junior high school. Schools were mainly classified as suburban (53%) or city (45.7%) and comprised of a relatively even split between high- (49.2%) and low-income (39.7%). Rates of AST progressively declined as distance thresholds increased from 59.10% (<¼ mile) to only 3.90% (≥1 miles). Over ¾ of the AST users in the sample lived within a reported distance of ½ mile from school. Descriptive results are displayed in Table 1.

Five environmental variables were significantly associated with AST in the initial model (intersection density, residential density, transit density, entertainment land use, and food-related land use) and were evaluated independently in multilevel models. When the interaction term with distance was considered, for every one-unit increase in intersection density, the odds of using AST were reduced among those living within ¼ mile of school (OR = 0.690, CI = 0.504, 0.941) but not among other distances. For every one-unit increase in food-related land use, the odds of using AST were reduced among those living within ¼ mile from school (OR = 0.563, CI = 0.562, 0.925) but not among other distance thresholds. No significant associations between AST and residential density, transit density, or entertainment land use were found. The results of the relationships between all five environmental variables and their interaction with distance on AST are displayed in Table 2.

As distance thresholds increased, the effect of intersections on the likelihood of walking and biking was reduced. Compared to those who lived ≥1 miles from school, the odds of AST as intersection density increased was reduced 31.0% using <¼-mile threshold; 18.86% using ¼–½-mile threshold; and 18.21% using <1-mile threshold. A similar effect was found with food-related land use. Compared to those who lived ≥1 miles from school, the odds of using AST as food-related land use increased was 59.4% for the ¼-mile threshold; 43.7% reduced odds for the ½-mile threshold; and 22.0% for the 1-mile threshold. Figure 1 and Figure 2 represent the predicted likelihood of AST based on low and high values of the environmental variables for <¼ mile and ≥1 miles, respectively. None of residential density, entertainment land use, and park land use was associated with the odds of using AST.

## 5. Discussion

This study considered the impact of distance on relationships between objectively measured features of the built environment and the likelihood of using active school transport (AST) modes, while controlling for student gender and grade, parent level of education, and school-level income and geographic locale. The results show that distance from home to school was a powerful predictor of behavior and may be a necessary consideration when evaluating how certain environmental features, such as intersection density and food-related land use, are related to walking and biking to school.

The effect of intersection density alone was not predictive of walking and biking to school in this sample, which aligns with other studies reporting null associations between intersection density and AST [19,22,31,44,45,46]. When distance was considered, a significant negative effect was found among those living within distance thresholds of <¼ compared to those living more than one mile, suggesting that the impact of intersections on AST may only be relevant when considered in conjunction with distance. Further, as distance cutoff thresholds expanded, the effect of intersection density on the odds of using AST was reduced, which supports the idea that programming and infrastructure improvements to overcome barriers related to the presence of intersections may be most effective if implemented within close distances (e.g., ¼ mile) to school sites. Parent perceptions of intersection danger may pertain to large street crossings [47] or time it takes to travel to school (i.e., exposure time), which are more likely to exist as distance increases. Thus, the success of programs such as park and walks, where coordinated points (e.g., park, parking lot) serve as locations for families to drop off students to be escorted the rest of the way to school on foot or bike, or walking school buses, where adult walkers escort children on the way to school, may be maximized near school sites with high intersection density.

Among this sample, food-related land use appeared as a barrier to walking and biking to school among those living within close distances (<¼ mile). It is possible that food outlets serve a proxy indicator of safety among children and parents. Food retail establishments may provide a steady flow of traffic, strangers coming and going, or a space for individuals to “hang out”, contributing to feelings of fear or social disorder [48]. Research on the relationship between AST and food-related land use characteristics has generally focused on whether nutrition [49] or body weight status [50,51,52] is affected by the presence of food outlets on the route to school; however, future research should examine the possibility that food stores could have safety implications related to transport behavior. Further, while safety is commonly acknowledged as a primary factor in general AST studies, the construct is rarely examined using GIS derived variables [14].

Residential density, alongside diversity, and design, is one of the frequently explored built-environmental determinants of travel behavior; however, in this study, residential density was not associated with odds of using AST. It may be that residential density and transit density, which was also not associated with AST in this sample, are more predictive of travel behavior among adults. While younger individuals may be affected by parks and greenspace or social interaction compared to environmental quality [53,54], land use dedicated to parks or entertainment was not related to walking in biking in this study. Park or green spaces may affect physical activity behaviors during non-school hours, which may have different predictors compared to AST. These null results may also be explained by the singular examination of density versus a collective examination of other related factors such as sidewalks, crosswalks, and connections between streets that have been linked to active travel behaviors among school children [29,55].

Across all models examined in this research, distance remained a powerful predictor of AST behavior in line with many other studies [14,26,27,28,29,30,31,32]. Improving school zoning policies and siting schools within neighborhoods is an ultimate goal to increase sustainable AST [56] as the median distance from home to school in the U.S. is 2.7 miles [57] and approximately 33% of U.S. students live within 1.5 miles of their school [58]. However, understanding where to implement interventions among those living within a reasonable distance is an important immediate strategy. Within our sample, almost half (45.4%) of AST users lived within a reported distance of ¼ mile from school and 80.8% lived within a reported ½ mile, supporting the notion that a walkable distance may realistically be quite less than the commonly used one-mile criterion.

There is a need to better understand where to target programming and infrastructure improvements aimed at overcoming environmental barriers to AST. These results suggest that strategies to mitigate the negative effects of intersections and food-related land use may be most effective in neighborhoods within ¼ mile from school. Focusing efforts toward improving the built environment around schools should be a collective priority as schools are critical community settings that have the potential to foster opportunities to develop lifelong health behaviors among children. Reducing environmental barriers in regions immediately surrounding school sites can contribute to increased rates of AST and affect other individual (e.g., perceptions of safety [59]) and psychosocial norms (e.g., auto-dependency [60]) surrounding walking and biking from a socioecological perspective [61].

### Limitations

While a strength of this study is the use of objective measurement of the home built environment, as subjective measurement may be associated with bias as active individuals are more likely to recognize how neighborhood features facilitate walking compared to those who use motorized forms of transport [14], this study has several limitations. First, because home location was determined by cross streets rather than exact home address, there is a potential for measurement error within the thresholds used to determine distance from school. In addition, distance and AST were self-reported, which may be another source of potential bias and error (e.g., recall and common source). The observed relationships with the food-related land use included all food outlet types. Future studies should examine the relationship by subgroups of outlets (e.g., limited service, full service, grocery, etc.). While outside of the scope of this research, it is important to note that a combination of built environment features may be important to consider to impact behavior change [62,63], and, in addition to the choice of mode of travel to school depending on environmental features, the decision to walk or bike likely depends on other social and cultural characteristics not examined in this study. Finally, we cannot determine causality as this research is cross-sectional.

## 6. Conclusions

Active school transport is a modifiable behavior that contributes to physical activity among children. This study suggests that certain built environment features may not directly impact behavior, but rather contribute to the likelihood of walking or biking within certain distance thresholds. While reducing distance between home and school remains a critical priority, there may be immediate potential for programming focused on overcoming barriers related to intersections and food outlets to increase rates of AST.

## Figures and Tables

**Figure 1 ijerph-17-07856-f001:**
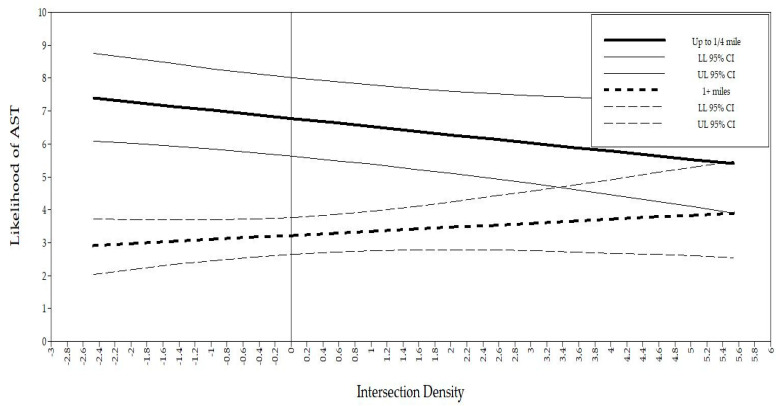
Interactions between intersection density and AST by distance.

**Figure 2 ijerph-17-07856-f002:**
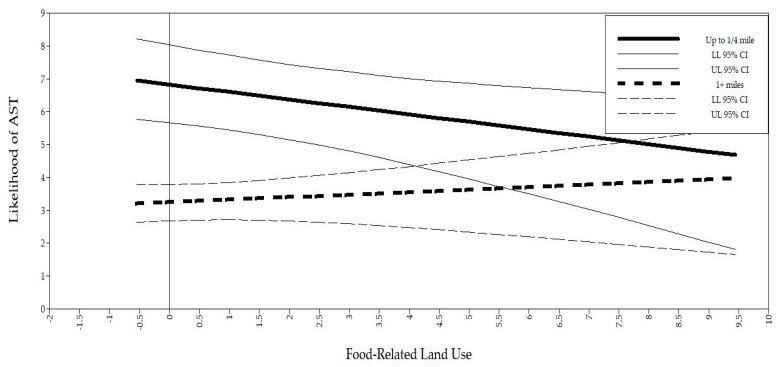
Interactions between food-related land use and AST by distance.

**Table 1 ijerph-17-07856-t001:** Descriptive information about sample and correlation with AST by reported distance (*n* = 2751).

Variable	All Distances	<¼ Mile (*n* = 594; 45.43% of All AST)	¼–½ Mile (*n* = 618; 35.35% of All AST)	½–1 Mile (*n* = 1549; 14.26% of All AST)	≥1 Miles (*n* = 983; 4.96% of All AST)	Correlation with AST ^a^
Gender						0.037
Boy	46.30%	45.70%	50.70%	49.70%	43.90%	
Girl	53.70%	54.30%	49.30%	50.30%	56.10%	
Grade						−0.050 *
3	25.80%	28.20%	23.60%	27.20%	24.50%	
4	26.50%	27.10%	29.30%	28.00%	25.00%	
5	22.70%	26.30%	25.10%	22.90%	18.20%	
6	11.60%	13.30%	12.60%	11.10%	10.50%	
7	9.20%	3.70%	4.90%	8.00%	15.20%	
8	4.20%	1.40%	4.40%	2.80%	6.70%	
Parent Education						19.727 **
Some HS or less	18.97%	20.20%	21.30%	18.70%	12.30%	
HS graduate	17.94%	19.70%	17.90%	16.50%	16.30%	
Some college ^1^	31.72%	35.00%	29.50%	30.40%	33.60%	
College graduate	31.37%	25.10%	31.10%	34.40%	37.80%	
Income						44.832 **
Low	34.10%	42.40%	36.20%	32.00%	23.40%	
Medium	14.80%	10.40%	14.30%	12.30%	18.30%	
High	51.20%	47.20%	49.50%	55.70%	58.30%	
Locale						7.120
City	49.05%	51.50%	43.10%	42.50%	48.20%	
Suburb	48.95%	47.80%	54.40%	56.10%	48.70%	
Other	2.00%	0.07%	2.50%	1.40%	3.20%	
Mode of arrival						
Walk or bike	27.20%	59.10%	44.20%	23.30%	3.90%	

** *p* < 0.01, * *p* < 0.05; ^1^ Includes Associates degree; ^a^ Gender association was evaluated with phi coefficient; grade was evaluated with biserial coefficient; all others were evaluated with chi-square statistic; and correlations were measured using whole sample; AST = active school transport (walking and biking combined). Frequencies ignore missing data so sample size varies from total *n.*

**Table 2 ijerph-17-07856-t002:** Fixed effects multilevel moderated logistic regression models predicting environmental variables on odds of AST by distance (n = 2751).

Environmental Variable	B	S.E.	Est./S.E.	OR	95% CI	95% CI
Intersection Density	0.122	0.126	0.971	1.130	0.935	1.399
Up to ¼ mile	**3.555**	**0.344**	**10.324**	**34.988**	**20.615**	**63.561**
¼ up to ½ mile	**2.962**	**0.314**	**9.423**	**19.337**	**12.158**	**33.920**
Up to 1 mile	**2.020**	**0.256**	**7.884**	**7.538**	**4.840**	**11.212**
Intersection Density × Up to ¼ mile *	**−0.371**	**0.190**	**−1.952**	**0.690**	**0.504**	**0.941**
Intersection Density × ¼ up to ½ mile *	−0.209	0.146	−1.430	0.811	0.623	1.002
Intersection Density × ½ up to 1 mile *	−0.201	0.169	−1.193	0.818	0.594	1.030
Residential Density	0.255	0.166	1.356	1.290	0.914	1.539
Up to ¼ mile	**3.553**	**0.341**	**10.423**	**34.918**	**20.573**	**62.178**
¼ up to ½ mile	**2.935**	**0.289**	**10.226**	**18.822**	**12.441**	**31.881**
Up to 1 mile	**2.007**	**0.255**	**7.874**	**7.441**	**4.816**	**11.101**
Residential Density × Up to ¼ mile *	−0.165	0.238	−0.695	0.848	0.582	1.251
Residential Density × ¼ up to ½ mile *	−0.343	0.188	−1.821	0.710	0.528	0.974
Residential Density × ½ up to 1 mile *	−0.012	0.245	−0.047	0.988	0.587	1.306
Transit Density	0.131	0.364	0.360	1.140	0.922	2.818
Up to ¼ mile	**3.590**	**0.344**	**10.447**	**36.234**	**20.369**	**61.930**
¼ up to ½ mile	**2.982**	**0.298**	**10.010**	**19.727**	**12.049**	**31.753**
Up to 1 mile	**2.034**	**0.246**	**8.281**	**7.645**	**4.938**	**10.924**
Transit Density × Up to ¼ mile *	−0.175	0.535	−0.327	0.839	0.230	1.191
Transit Density × ¼ up to ½ mile *	−0.146	0.499	−0.292	0.864	0.258	1.130
Transit Density × ½ up to 1 mile *	−0.285	0.436	−0.654	0.752	0.267	1.108
Entertainment Land Use	0.289	1.608	0.180	1.335	0.861	1.972
Up to ¼ mile	**3.569**	**0.521**	**6.850**	**18.672**	**20.512**	**64.715**
¼ up to ½ mile	**2.961**	**0.495**	**5.981**	**19.317**	**11.989**	**33.082**
Up to 1 mile	**2.023**	**0.582**	**3.474**	**7.561**	**4.702**	**11.496**
Entertainment Land Use × Up to ¼ mile *	−0.233	1.650	−0.141	0.792	0.422	1.226
Entertainment Land Use × ¼ up to ½ mile *	−0.279	1.678	−0.166	0.757	0.464	1.165
Entertainment Land Use × ½ up to 1 mile *	−0.271	2.226	−0.122	0.763	0.389	1.267
Food Land Use	0.077	0.416	0.184	1.080	0.910	1.221
Up to ¼ mile	**3.579**	**0.408**	**8.767**	**19.144**	**20.739**	**63.118**
¼ up to ½ mile	**2.986**	**0.380**	**7.855**	**0.722**	**12.207**	**33.920**
Up to 1 mile	**2.063**	**0.342**	**6.033**	**0.406**	**4.978**	**11.763**
Food Land Use × Up to ¼ mile *	**−0.304**	**0.432**	**−0.703**	**0.563**	**0.562**	**0.925**
Food Land Use × ¼ up to ½ mile *	−0.247	0.447	−0.554	0.780	0.575	1.096
Food Land Use × ½ up to 1 mile *	−0.221	0.446	−0.496	0.802	0.592	1.062

* Reference category = 1 + mile distances; *Note*. bold values significant at *p* < 0.05.

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
