# Peer review of "The Moderating Effect of Distance on Features of the Built Environment and Active School Transport"

_ijerph, 2020, doi:10.3390/ijerph17217856_

Round 1
Reviewer 1 Report
The current study investigated the relationship between riding or walking to school, the bulit environment and distance from home to school in a large sample. The results were mildly surprising and have a real world application.
a few points to note
- food-related land use needs to be defined as early as possible as it is an ambiguous phrase that could be somewhat confusing without clarification
- table 2 is somewhat busy and difficult to understand. would be better with improved formatting and section headings
- Figure 1 is very small and would also be improved with more detail and better formatting
- I think there are a number of other interpretations that have not been explored as to why food retailers might be associated with decreased willingness to active transport (e.g. they tend to be on main roads)
- bottom of page 8 - last paragraph, last sentence - maybe a typo but very hard to tell what is meant here
- i think the first sentence of the conclusion can be stronger safely - just say "...that contributes to physical activity..."
Author Response
Thank you for the helpful review. Please see our response in the attached document.

Reviewer 2 Report
This study explored the interaction between objectively measured features of the built environment and distance from home to school to determine which environmental features are associated with AST by distance categories.
The research theme is important, but there are unclear points in the analysis part. Additional description (and corrections if necessary) is required.
Major comments:
- How did you calculate the correlation with categorical variables such as "parent education" in Table 1?
- Is "distance" in Table 2 a continuous variable? What is the difference between "Intersection Density x Distance Criterion" and "Intersection Density x <Distance"? This table is most important in this article, but lacks the necessary notes.
- I tried to calculate the CI from B and SE in Table 2, but it didn't match your result.
- Please add a description as to which values in Table 2 were used to create Figure 1.
- You wrote "while younger individuals may be affected by parks and greenspace", but wasn't "park categories" entered into the first model and non-significant? The non-significant environment variables should also be mentioned in Discussions.
Minor comments:
- While there is no clear explanation, you used the random intercept model, right?
- I didn't know where in the table the following content could be read. “Schools were mainly classified as suburban (53%) or city (45.7%) and comprised of a relatively even split between high- (49.2%) and low-income (39.7%).” (p.3)
- Isn't "59.3%" in the first line of page 4 "53.3%"?
- How did you calculate 52.70%, 77.78%, 94.29% in the first row of Table 1? For example, 27.2% of "All distances" (n = 2,751), or about 748, seem to be AST users. On the other hand, within less than 1/4 mile (n = 646), 59.1% or about 382 are AST users. 382 divided by 748 is 51.0% and do not match with the figure in the first row.
- "Income" in Table 1 should be placed on the left side.
- It seems that significant parts are bold in Table 2, but there seem to be a few mistakes.
- In the PDF I downloaded, the figures overlap and I can't see the legend. (Figure 1)
- You wrote "food-related land use may be most effective in neighborhoods within ¼ or ½ mile from school" (p.9), but "Food Land Use x <Distance" was also significant within 1-mile distance criterion.
Author Response

(The authors gave the same response as above.)
